# Functional dynamic prosthesis alignment maintained across varying footwear using a modular prosthetic ankle-feet system

**Nicole R. Walker**[1,2]*, **Myrriah P. Laine Dyreson**[1,2◉], **Juan E. Cave II**[1◉], **Kali R. Mansur**[1◉], **Kelly J. Yun**[1◉], **John M. Looft**[1,2◉], **Andrew H. Hansen**[1,2,3◉]

**1** Rehabilitation and Engineering Center for Optimizing Veteran Engagement and Reintegration (RECOVER), Minneapolis VA Health Care System, Minneapolis, Minnesota, United States of America, **2** Department of Family Medicine and Community Health, University of Minnesota – Twin Cities, Minneapolis, Minnesota, United States of America, **3** Department of Biomedical Engineering, University of Minnesota – Twin Cities, Minneapolis, Minnesota, United States of America

◉ These authors contributed equally to this work.

* Nicole.Walker6@va.gov

## Abstract

Changing footwear often presents a challenge for lower extremity prosthesis users. When prosthesis alignment is completed by the Certified Prosthetist, the prosthetic foot is set at an angle accommodating a single shoe heel rise (shoe heel height minus forefoot height); deviation from this heel rise causes misalignment of the prosthesis. To address this problem, the Rehabilitation & Engineering Center for Optimizing Veteran Engagement & Reintegration (RECOVER) has developed a modular ankle-feet system allowing for the use of footwear of varying heel rises without the need for realignment by the prosthesis user. The primary aim of this study was to understand if clinically acceptable prosthesis alignment is maintained as prosthesis users change between modular foot-shoe sets. Three women transtibial prosthesis users self-selected three pairs of footwear with a heel rise up to 10 cm. Using the modular prosthetic ankle-feet system, participants completed five walking trials per foot-shoe set in the motion analysis laboratory. Reflective markers were used to track the location of the prosthetic socket during walking. These data in combination with center of pressure measurements were used to calculate ankle-foot-shoe rollover shapes and the location of the origin of the best-fit circle for each rollover shape. The locations of the resulting best-fit circle origins indicate prosthesis alignment is maintained within clinically acceptable parameters as users change between foot-shoe sets. These findings have implications for improving footwear options for people with lower-extremity amputations.

**Data availability statement:** All relevant data are within the paper and its Supporting Information files.

**Funding:** This work was supported by two Merit Review Awards from the Rehabilitation Research and Development Service of the United States Department of Veterans Affairs (https://www.research.va.gov/), award numbers RX002634 and RX004256, awarded to Dr. Andrew Hansen. This work was also supported by a Graduate Assistantship from the University of Minnesota Informatics Institute MnDRIVE program (https://research.umn.edu/industry-partnership/mndrive), award number UMII-GA-0866719453, awarded to Ms. Nicole Walker. Funders were not involved in the study design, in the collection, analysis and interpretation of data; in the writing of the manuscript; or in the decision to submit the manuscript for publication.

**Competing interests:** I have read the journal's policy and the authors of this manuscript have the following competing interests: Dr. Andrew Hansen is an inventor on a pending patent related to the modular prosthetic ankle-feet system described in this project. The pending patent is owned by the United States Department of Veterans Affairs.

## Introduction

Using footwear of varying heel rise (heel height minus forefoot height) is a challenge for individuals who use lower extremity prostheses. In individuals without leg amputation, the ankle-foot complex adapts to shoes of various heel rises, plantarflexing to accommodate the heel rise of the shoe [1]. However, individuals with amputations who use prosthetic feet have no natural adaptation to shoes of different heel rises. When aligned by the prosthetist, the prosthetic ankle is set at an angle to accommodate a single shoe heel rise [2]. Placing the aligned prosthetic foot in a different shoe with a heel rise that is either lower or higher causes misalignment of the prosthesis. Use of a misaligned prosthesis has potential consequences for the user, including pain in the residual limb and remaining joints, damage to skin, or increased risk of experiencing a fall [3].

The Rehabilitation & Engineering Center for Optimizing Veteran Engagement & Reintegration (RECOVER) at the Minneapolis VA Health Care System has developed a modular ankle-feet system that allows for the fabrication of a custom foot shape designed to match nearly any pair of shoes (Fig 1) [4]. The modular ankle system uses additive manufacturing to create a custom foot shape matching the contours and size of any shoe with a heel height up to four inches (10 cm). The foot shape and shoe become a set, and each set interfaces with a multiaxial ankle unit. Resistance in each plane is tuned to the user via silicone rubber bumpers and bushings, and the alignment of the prosthetic ankle is set by the prosthetist. For the system to function properly, prosthesis alignment must be preserved across a variety of foot-shoe sets, allowing the prosthesis user to repeatedly switch foot-shoe sets without the need for realignment by the prosthetist.

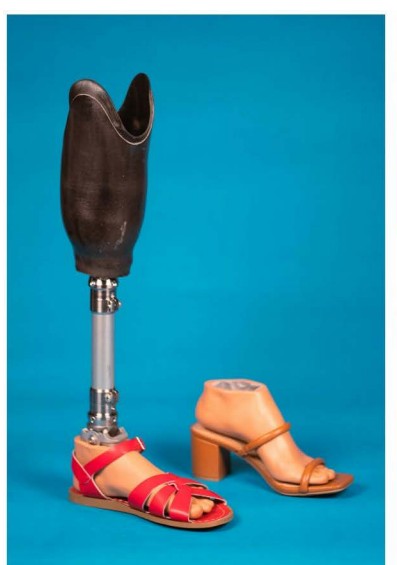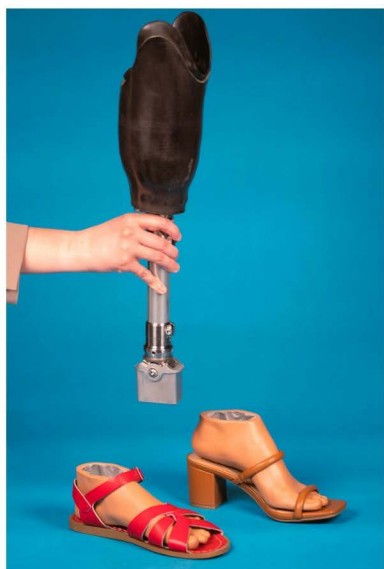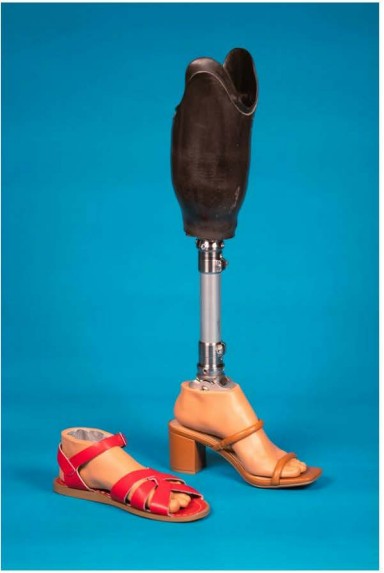

**Fig 1. RECOVER modular prosthetic ankle-feet system.**

One method for understanding preservation of alignment across foot-shoe sets is to measure the ankle-foot-shoe rollover shape achieved during walking. The ankle-foot-shoe rollover shape represents the effective rocker shape in the sagittal plane that the ankle, foot, and shoe create during walking (Fig 2). Ankle-foot rollover shapes resemble and can be characterized with the lower arc of a circle. When applying this concept to prosthetic foot alignment, Hansen et al. hypothesize that Certified Prosthetists progressively align the prosthesis toward an "ideal" rollover shape, such as the rollover shape created by the anatomical foot and ankle complex. As the prosthesis alignment is changed, the arcs of the rollover shapes become gradually more similar to the "ideal" shape until the Certified Prosthetist and patient are satisfied with the prosthesis alignment. Similarly, the locations of the center of the rollover arcs in the sagittal plane begin to move toward the "ideal" location [5]. Rollover shape, radii, and arc center location are sensitive to varying characteristics of the prosthetic foot and alignment, including plantarflexion and dorsiflexion [6], anterior and posterior translation of the foot relative to the prosthetic socket, [7] ankle range of motion [8], and keel stiffness [9].

When neurotypical humans walk with a series of shoes of different heel rises, the x-coordinate location of the center of the arc ($x_0$), representing anterior-posterior shift, does not change appreciably, and the y-coordinate of the center of the arc ($y_0$) varies in congruence with the heel rise of the shoe. This preservation of the $x_0$ location is a result of the ability of the anatomical ankle and foot complex to adapt to changes in heel height to maintain gait mechanics [10]. However, when considering lower extremity prosthesis users, the prosthetic foot does not automatically adapt to changes in heel height. Even slight changes to the heel height of the shoe result in measurable changes to the rollover shape and arc center location during walking. A 2009 study of rollover shapes during simulated walking with prosthetic feet reported large differences in the anterior-posterior shift of the center of the arc when changing between shoes of varying heel heights, indicating the sensitivity of the $x_0$ shift to heel height [6]. The Certified Prosthetist may realign the prosthetic foot to accommodate changes in heel height, but this would need to be done every time the user changed heel heights to maintain the proper rollover shape and arc center location across footwear.

Further, the clinically acceptable variation in the location of $x_0$ while changing prosthetic feet has not been well-studied. Zahedi et al. previously quantified variations in prosthesis alignment in the x-direction when a single Certified Prosthetist aligned and realigned a single transtibial prosthesis multiple times for the same user to the prosthetist's and user's satisfaction [3]. Over the course of two years and 19 realignments, the range of anterior-posterior translations of the socket relative to the foot was 1.6 cm. Thus, this range of anterior translation of the socket relative to the foot may be considered

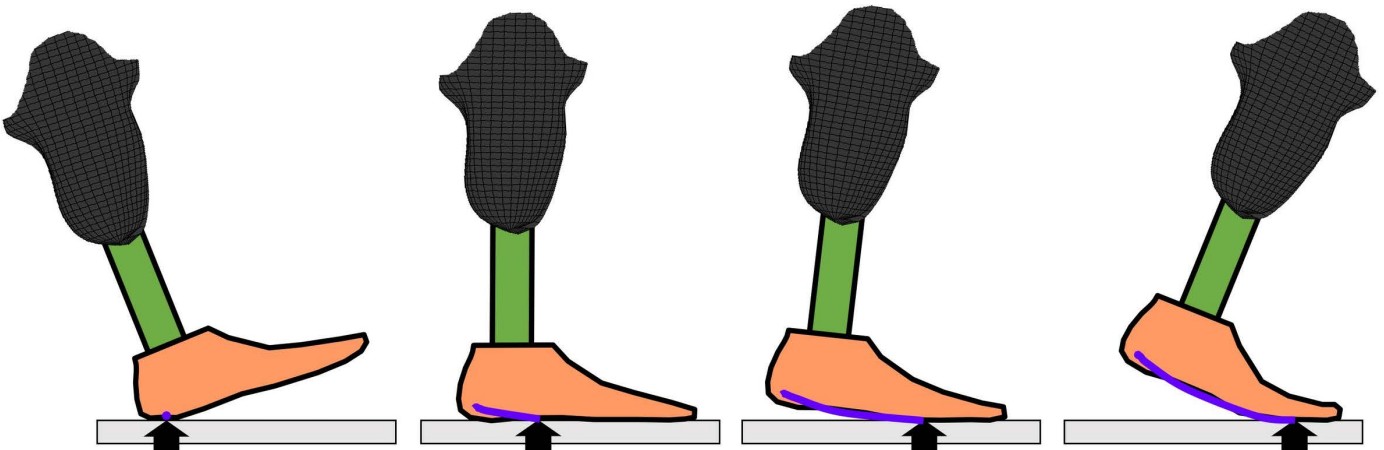

**Fig 2. Ankle-feet rollover shape development.** Rollover shapes are created by transforming the center of pressure trajectory (measured by laboratory-based force plates) to a socket-based coordinate system using markers placed on the transtibial prosthetic socket. The rollover shape represents the effective rocker shape the ankle-foot system conforms to during the stance phase of walking.

the clinically acceptable range of prosthesis alignment. The RECOVER ankle-feet system is designed to maintain the prosthetic alignment without adjustment from a prosthetist or the user as foot-shoe sets are changed. The goal of this pilot study was to conduct a biomechanical analysis of lower extremity prosthesis users walking with the novel RECOVER ankle-feet system to evaluate the preservation of alignment between prosthetic foot-shoe sets. We hypothesized that prosthesis alignment in the sagittal plane, specifically the anterior-posterior shift of rollover shapes, would not vary beyond the clinically acceptable parameter reported by Zahedi across three different foot-shoe conditions.

## Methods

### Study procedures

This study employed a single-arm, randomized crossover design. Recruited participants trialed three different foot-shoe sets, including three 3D-printed foot shapes designed to fit their selected footwear. Each participant's clinically prescribed prosthetic foot was removed maintaining alignment by the study Certified Prosthetist (author NRW). The modular ankle-feet system was attached to the participant's clinical prosthetic socket and aligned to an alignment condition satisfactory to the prosthetist and user for all three sets, replicating the process used in clinical settings. Participants were allowed to walk in the lab until they were comfortable in the shoes and satisfied with the prosthesis alignment. Biomechanical motion data were collected for each participant using all three foot-shoe sets. Following motion capture, the participant's pre-scribed prosthetic foot was replaced and secured by the study prosthetist, and their participation in the study concluded.

### Motion analysis

The motion analysis laboratory is outfitted with 20 Qualisys (Qualisys AB; Göteborg, Sweden) motion capture cameras, two Qualisys Miqus high-speed video cameras, and six AMTI (Advanced Mechanical Technology, Inc.; Watertown, Massachusetts) Optima floor-embedded force plates. Twenty-five reflective markers in a custom configuration were used to track the prosthesis and non-amputated sides; seven of these markers were used to track the prosthetic socket-based coordinate frame which is crucial to measuring ankle-foot-shoe rollover shape (Fig 3). Motion data were collected using all three

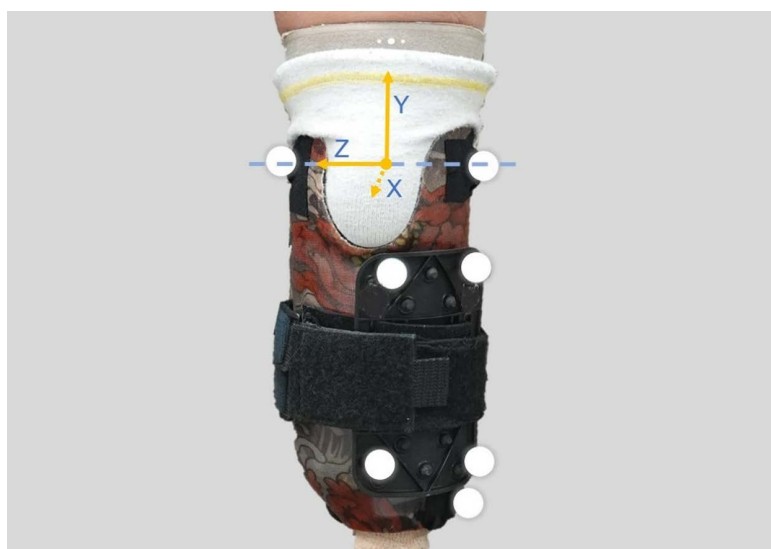

**Fig 3. Prosthetic socket reference frame.** The y-axis indicates vertical, the z-axis horizontal, and the x-axis the line of forward progression. The marker cluster is located on the lateral side of the prosthetic socket.

foot-shoe sets in a randomized order, and each walking trial continued until five clean force plate strikes were achieved for the prosthesis side.

Marker trajectories for each walking trial were mapped using Qualisys Track Manager (v2023.1). Walking trials were trimmed and selected to include only five clean force plate strikes per side in each foot-shoe set. Anatomical segment modeling and motion calculations were conducted in Visual3D (HAS-Motion; Ontario, Canada; v2021.11.3). Rollover shape was measured as the center of pressure progression in the socket-based coordinate frame [11]. Best-fit circular arcs were calculated for each rollover shape using a previously described technique, yielding a best-fit radius (R) and location of the best-fit arc center $(x_0, y_0)$ [12]. All calculations were completed in Python (Python Software Foundation; Beaverton, Oregon; v3.11.9) using custom code developed by authors AHH and MPLD. Raw rollover shape data are provided in Supporting Information File 1.

### Hypothesis testing

The central hypothesis posited that prosthesis alignment would not vary beyond a clinically acceptable parameter as users of the modular ankle-feet system switched between foot-shoe sets. To evaluate this hypothesis, the ankle-foot-shoe rollover shapes for each of the five walking trials per condition were plotted. Rollover shape radius is coupled with the $x_0$ coordinate of the center of the circular arc [13]. To avoid the effects of this coupling on the location of $x_0$, the radius of the best fit circular arc was standardized within each participant by using the median radius across the fifteen walking trials (five trials per foot-shoe condition). Best-fit circular arc locations were then found using this standardized radius as described earlier [12]. The centers $(x_0, y_0)$ of the best-fit circular arcs for all fifteen walking trials were plotted for each participant. When considering the range of $x_0$ locations per participant, we accepted our hypothesis if the range was less than or equal to 1.6 cm and rejected the hypothesis if the range was larger than 1.6 cm.

### Pilot testing results

#### Participants

The Minneapolis VA Health Care System Institutional Review Board (IRB) approved this study and written informed consent to participate was collected (IRB #1697326). Participant recruitment began 1 March 2021, and concluded 31 October 2023. Three women with transtibial amputations participated in biomechanical testing. Participants were 38 years old on average, with a range from 29 to 47 years at visit date. On average, participants had experienced their amputation 15.4 years prior to their visit, with a range from 4 to 36 years since amputation. All participants were regular prosthesis users with well-fitting sockets and no current skin or residual limb concerns. Additional participant demographics are included in Supporting Information File 2. Each participant self-selected three pairs of shoes in varying heel rises to trial the modular ankle feet system. Participant selected shoes and heel rise measurements are displayed in Fig 4.

| | Participant 1 | | | Participant 2 | | | Participant 3 | | |
|---|---|---|---|---|---|---|---|---|---|
| Shoe Style | Sandal | Ankle Boot | Stiletto Heel | Kitten Heel | Wedge Sneaker | Ankle Boot | Ankle Boot | Heel Sandal | Stiletto Heel |
| Heel Rise (cm) | 2 | 3 | 10 | 2 | 4 | 5 | 2 | 5 | 10 |

**Fig 4. Participant selected footwear.** Participant selected footwear, shoe style, and heel rise for all shoes included in the study.

## Rollover shape and prosthesis alignment

Rollover shapes were plotted per participant for each walking trial (Fig 5). Rollover shape locations changed in the y-direction as anticipated based on shoe heel rise. Additionally, the rollover shapes run relatively parallel to one another, with no appreciable change in shape or angulation across trials. To better understand the preservation of alignment, the $x_0$ locations of the rollover shape arc of best fit origin were calculated and are displayed in Fig 6. For all participants, the range of $x_0$ translation of the arc of best fit origin across foot-shoe sets varied less than or equal to 1.6 cm; participant 1 $x_0$ locations varied by 1.3 cm, participant 2 by 0.7 cm, and participant 3 by 1.6 cm.

## Discussion

The primary objective of this pilot test was to understand if prosthetic foot alignment could be maintained during walking using the [research program] modular prosthetic ankle feet system and footwear of varying heel heights. The findings of this pilot test supported the hypothesis that the RECOVER modular prosthetic ankle-feet system preserves prosthesis alignment across foot-shoe sets of varying heel height. Because all other influential prosthetic foot variables were maintained (e.g., ankle range of motion, keel stiffness) across foot-shoe sets and did not appreciably change rollover shape characteristics, users of the system should be able to swap foot-shoe sets of varying heel height without the need for prosthesis realignment.

To the authors' knowledge, this pilot study presents the first use of Zahedi et al.'s measurements of acceptable socket translation to understand the preservation of prosthesis alignment between prosthetic feet systems. Zahedi et al. have proposed 1.6 cm as an acceptable range of posterior-anterior translation of the prosthetic foot resulting in alignment that is satisfactory for both the Certified Prosthetist and prosthesis user [3]. The range of anterior-posterior translation of the $x_0$ location was less than or equal to 1.6 cm for all three participants across the selected foot-shoe sets, indicating clinically acceptable alignment in this plane.

A number of heel height adjustable prosthetic feet are available on the market which allow the prosthesis user to adapt their alignment for different footwear. However, the difficulty of consistently or accurately realigning one's own prosthesis is supported by previous literature. A study reported by Kent et al. indicated inconsistent changes in ankle plantarflexion and dorsiflexion by prosthesis users when accommodating flat or heeled shoes compared to the alignment achieved at baseline by a Certified Prosthetist. This finding indicates that users may not be skilled enough to accurately adjust their ankle alignment compared to Certified Prosthetists [14]. Hydraulic prosthetic ankles, including those with microprocessor-controlled resistance, may also accommodate slight changes in heel height. Increases in heel height are limited by the available range of motion of these feet; increasing heel height reduces the available amount of plantarflexion range at initial contact, which may negatively influence walking mechanics for the prosthesis user [4]. The RECOVER modular prosthetic ankle-feet system eliminates these problems by requiring the user to only change the 3D printed foot shape when changing shoes; the ankle alignment is set by the prosthetist and does not change based on the use of different foot-shoe sets, and ankle range of motion is inherent to the multiaxial ankle unit and identical without regard to which foot-shoe set is used.

A limitation of the work is our investigation of alignment in the sagittal plane only. Future work could examine coronal and transverse plane alignment measures using three-dimensional rollover surface measures. However, the participants and prosthetists did not note any issues with either the coronal or transverse plane alignments in this study. This study is also limited by a small number of participants; however, this group of participants was sufficient to demonstrate the methodology for examining similarities in sagittal plane alignment. Future work will use this methodology to examine sagittal plane alignment in a larger cohort of participants. Prosthesis users may be averse to using the modular ankle-feet system due to needing a unique foot shape for each pair of shoes. There may be some flexibility in this approach—e.g., a foot shape designed for a specific shoe with a three-inch heel may also accommodate additional shoes with the same heel

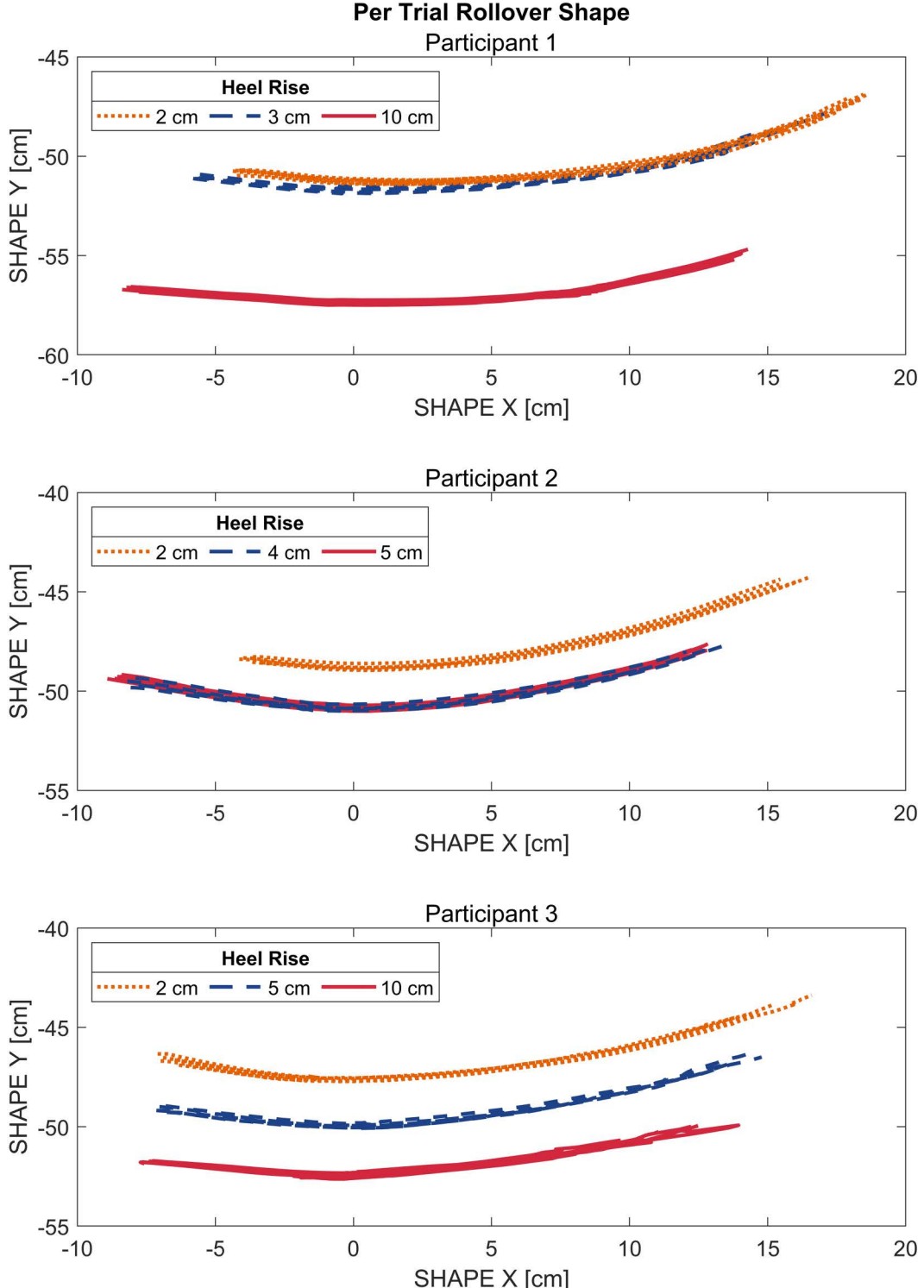

**Fig 5. Per trial rollover shapes.** Per trial rollover shapes plotted per participant. Changes in the vertical y-direction correspond to changes in heel rise. All rollover shapes run relatively parallel to each other with no appreciable change in shape or angulation per trial, indicating preservation of angular alignment in the sagittal plane between foot-shoe combinations.

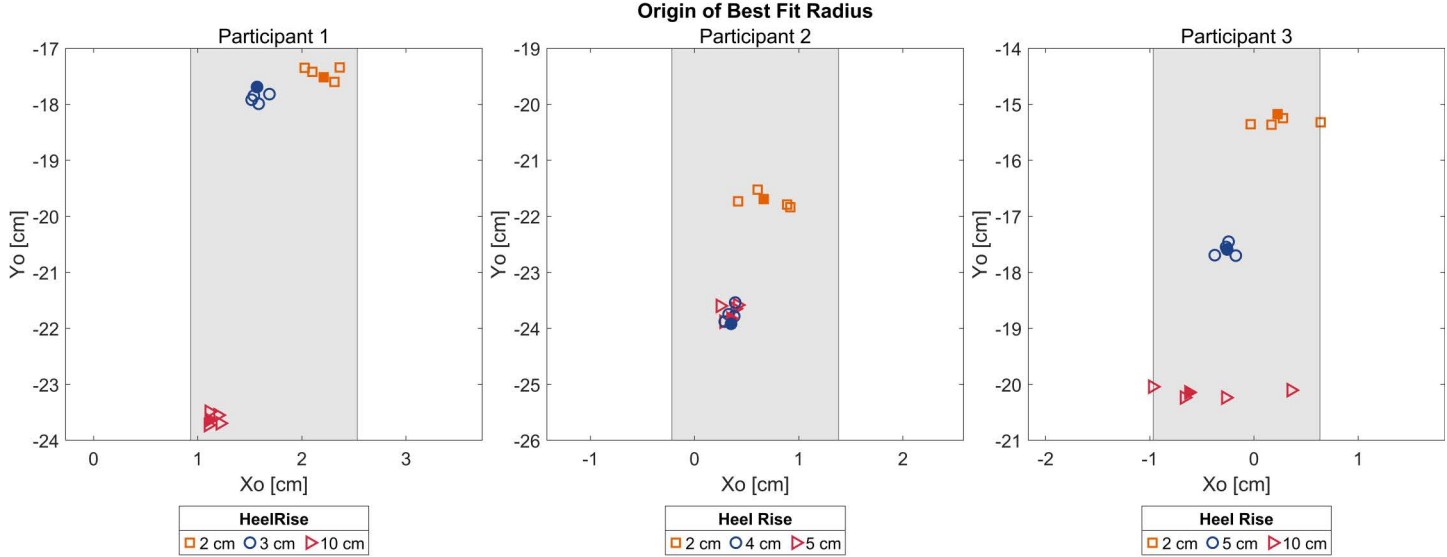

**Fig 6. Origins of best fit radius.** Per trial plots of location of origin of the best-fit circle per participant. The shaded band represents a range of 1.6 cm, the proposed clinically acceptable range for alignment variation in the x-direction. For all participants, the range of x0 translation of the arc of best fit origin across foot-shoe combinations varied by equal to or less than 1.6 cm.

rise—but ultimately users will require many custom foot shapes. Some prosthesis users already use a variety of prosthetic feet for different applications and may be comfortable with this aspect of using the modular ankle-feet system, but this limitation should be considered on an individual basis. Finally, prosthetic feet characteristics and features have changed significantly since Zahedi's 1986 publication on acceptable anterior-posterior shift. Thus, there is a chance that the 1.6 cm shift used to validate the findings of this pilot work may no longer accurately represent the acceptable range for modern prosthetic feet; further testing using current technologies would offer insight into the usefulness of this value for validating rollover shape metrics.

In conclusion, this work demonstrates a new methodology for examining the similarities in functional dynamic alignment of prostheses using rollover shape and variability in alignments measured previously by Zahedi et al. This methodology was used in a pilot study to show similarities in alignment of a new modular prosthetic ankle-feet system with implications for improving footwear options for persons with amputations.

## Supporting information

**S1 Information File. Raw Rollover Shape Data.**
(XLSX)

**S2 Information File. Table of Additional Participant Demographics.**
(DOCX)

## Author contributions

**Conceptualization:** Nicole R. Walker, Andrew H. Hansen.

**Data curation:** Myrriah P. Laine Dyreson.

**Formal analysis:** Myrriah P. Laine Dyreson.

**Funding acquisition:** Andrew H. Hansen.

**Investigation:** Nicole R. Walker, Juan E. Cave II, Kali R. Mansur, Kelly J. Yun.

**Methodology:** Nicole R. Walker, Myrriah P. Laine Dyreson, John M. Looft, Andrew H. Hansen.

**Project administration:** Nicole R. Walker, Kali R. Mansur, Kelly J. Yun.

**Software:** Myrriah P. Laine Dyreson, Andrew H. Hansen.

**Visualization:** Nicole R. Walker, Myrriah P. Laine Dyreson.

**Writing – original draft:** Nicole R. Walker, Andrew H. Hansen.

**Writing – review & editing:** Nicole R. Walker, Myrriah P. Laine Dyreson, Juan E. Cave II, Kali R. Mansur, Kelly J. Yun, John M. Looft, Andrew H. Hansen.

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
