## [Decision Letter · Decision Letter 0]

22 Aug 2024

PONE-D-24-25730Functional dynamic prosthesis alignment maintained across varying footwear using a modular prosthetic ankle-feet systemPLOS ONE

Dear Dr. Walker,

Thank you for submitting your manuscript to PLOS ONE. After careful consideration, we feel that it has merit but does not fully meet PLOS ONE’s publication criteria as it currently stands. Therefore, we invite you to submit a revised version of the manuscript that addresses the points raised during the review process.

We look forward to receiving your revised manuscript.

Kind regards,

Arezoo Eshraghi, Ph.D.

Academic Editor

PLOS ONE

Journal Requirements:

2. Thank you for stating the following in the Competing Interests section: "I have read the journal's policy and the authors of this manuscript have the following competing interests: Dr. Andrew Hansen is an inventor on a pending patent related to the modular prosthetic ankle-feet system described in this project. The pending patent is owned by the United States Department of Veterans Affairs."

Reviewers' comments:

Reviewer's Responses to Questions

**Comments to the Author**

1. Is the manuscript technically sound, and do the data support the conclusions?

Reviewer #1: Partly

2. Has the statistical analysis been performed appropriately and rigorously? 

Reviewer #1: I Don't Know

3. Have the authors made all data underlying the findings in their manuscript fully available?

Reviewer #1: Yes

4. Is the manuscript presented in an intelligible fashion and written in standard English?

Reviewer #1: Yes

5. Review Comments to the Author

Reviewer #1: This paper assessed if the RECOVER modular prosthetic ankle-feet system can maintain sagittal plane alignment when prosthesis users change their shoes, each having different heel heights. The researchers completed this by attaching the RECOVER system to the participant’s socket, have the participant wear three different pairs of shoes with varying heel heights, and collect motion capture and force plate data to analyze roll-over shape to determine if the anterior-posterior alignment was maintained. I have split my review into major and minor comments.

Major comments:

The authors specifically investigate the anterior posterior shift of the prosthesis user’s rollover shape. There is no information stating why the anterior posterior shift is important and should be researched. The authors need to add a paragraph explaining why this is an important metric to investigate, including but not limited to what happens if the anterior posterior shift is too drastic.

One of the main points being made in the paper is that their system does not dramatically change rollover shape. However, it is not clear how sensitive roll over shape is to relevant variables in general (alignment, stiffness, etc.). For instance, is it possible that changes to prosthesis stiffness or alignment would similarly not impact roll-over shape, even though we know that they have other different effects on gait? Overall, the argument that rollover shape is little changed and thus the device works, is missing validation of the sensitivity of rollover shape to detect meaningful differences in gait / prosthesis mechanics. This should be addressed through an explicit comparison with other variables in prior published literature.

The majority of the needed information is not directly in the paper but rather only found in the references which made reading this challenging. The RECOVER ankle-feet system is only briefly explained without enough information to know exactly how the system works. The authors need to add this information into this paper and not just rely on the reference. Furthermore, roll-over shape is not an outcome metric that is easy for most people to understand. Since roll-over shape is the main outcome measure, the authors need to add sufficient detail in the paper about what it is and how it is found, not just relying on the references.

Prosthetic feet have changed measurably since 1986, therefore, it may not be correct to assume that the clinically acceptable parameter of 1.6 cm found in the Zahedi paper would hold true for prosthesis users today. The prosthesis users in the Zahedi paper only used Sach or uniaxial prosthetic feet from back then which cannot be fairly compared to newer flexible keel feet for those with the functional level of K2 or the energy storage and return (ESR) feet used by those with the functional level of K3. This may also hold true for the sockets, liners, and suspension that were used in the 1980’s compared to today’s technology. The authors may not be able to change this issue for this current publication but may be something to be aware of for their future research.

Minor comments:

The authors mention that there are “several products available that allow prosthesis users to adapt their alignment for different footwear” but then alignment becomes the focus, specifically how patients have a hard time aligning their own prosthesis correctly. Products and adjusting prosthesis alignment are different unless the product is a wrench. The authors should add a section explaining hydraulic feet, what these products are lacking, and how their system may be better for prosthetic users.

For proper use of the RECOVER ankle-feet system, the prosthesis user would need a specific 3D printed foot for each of their shoes that have a unique heel height (in other words, multiple prosthetic feet). Some prosthesis users will be fine with this and some will not. The authors should consider mentioning this as a possible limitation.

The authors should add more references pertaining to current prosthetic feet used in clinic, alignment, biomechanics of prosthesis users, the impacts alignment can have on the users, and rollover shape.

6. PLOS authors have the option to publish the peer review history of their article (what does this mean? ). If published, this will include your full peer review and any attached files.

**Do you want your identity to be public for this peer review?** For information about this choice, including consent withdrawal, please see our Privacy Policy .

Reviewer #1: No

---

## [Author Response · Author response to Decision Letter 1]

12 Dec 2024

We greatly appreciate the reviewer’s thoughtful evaluation of our manuscript. Please see our responses to the

reviewer.

Reviewer 1:

“The authors specifically investigate the anterior posterior shift of the prosthesis user’s rollover shape. There is

no information stating why the anterior posterior shift is important and should be researched. The authors need

to add a paragraph explaining why this is an important metric to investigate, including but not limited to what

happens if the anterior posterior shift is too drastic.”

Additional information and citations have been added to the introduction to further describe rollover shape and

the anterior posterior shift (referred to as the x0 location of the center of the arc). Additionally, further

information has been included to describe the influence of heel height on the x0 location, the primary variable of

interest for this pilot work. See lines 48-58 and 65-75 in the tracked changes manuscript.

“One of the main points being made in the paper is that their system does not dramatically change rollover

shape. However, it is not clear how sensitive roll over shape is to relevant variables in general (alignment,

stiffness, etc.). For instance, is it possible that changes to prosthesis stiffness or alignment would similarly not

impact roll-over shape, even though we know that they have other different effects on gait? Overall, the

argument that rollover shape is little changed and thus the device works, is missing validation of the sensitivity

of rollover shape to detect meaningful differences in gait / prosthesis mechanics. This should be addressed

through an explicit comparison with other variables in prior published literature.”

Citations reporting the sensitivity of rollover shape (and more specifically of the x0 location) have been added in

lines 55-58 in the tracked changes manuscript.

“The majority of the needed information is not directly in the paper but rather only found in the references

which made reading this challenging. The RECOVER ankle-feet system is only briefly explained without

enough information to know exactly how the system works. The authors need to add this information into this

paper and not just rely on the reference. Furthermore, roll-over shape is not an outcome metric that is easy for

most people to understand. Since rollover shape is the main outcome measure, the authors need to add

sufficient detail in the paper about what it is and how it is found, not just relying on the references.”

Additional detail describing the RECOVER ankle-feet system has been added in line 34-42 of the tracked

changes manuscript. An additional figure (Figure 2) has been added to illustrate the ankle-foot rollover shape

as it applies to measuring this socket based metric, and additional information about rollover shape and arc

center location have been added in lines 48-58 and 65-75 in the tracked changes manuscript.

“Prosthetic feet have changed measurably since 1986, therefore, it may not be correct to assume that the

clinically acceptable parameter of 1.6 cm found in the Zahedi paper would hold true for prosthesis users today.

The prosthesis users in the Zahedi paper only used Sach or uniaxial prosthetic feet from back then which

cannot be fairly compared to newer flexible keel feet for those with the functional level of K2 or the energy

storage and return (ESR) feet used by those with the functional level of K3. This may also hold true for the

sockets, liners, and suspension that were used in the 1980’s compared to today’s technology. The authors may

not be able to change this issue for this current publication but may be something to be aware of for their future

research.”

The authors certainly acknowledge that prosthetic feet technology has changed considerably since Zahedi’s

publication in 1986. However, this publication remains (to our knowledge) the most meaningful piece of

literature exploring the clinically acceptable variability of anterior-posterior alignment among transtibial

prosthesis users. An acknowledgment of the potential limitation of the age of Zahedi’s publication has been

added to the Discussion section in lines 217-223 in the tracked changes manuscript.

“The authors mention that there are “several products available that allow prosthesis users to adapt their

alignment for different footwear” but then alignment becomes the focus, specifically how patients have a hard

time aligning their own prosthesis correctly. Products and adjusting prosthesis alignment are different unless

the product is a wrench. The authors should add a section explaining hydraulic feet, what these products are

lacking, and how their system may be better for prosthetic users.”

Additional information about currently available prosthetic feet technologies, including heel height adjustable

feet and feet with hydraulic or microprocessor-controlled ankles, and their adaptability to heel height has been

added to the discussion section in lines 187-189 and lines 194-198.

“For proper use of the RECOVER ankle-feet system, the prosthesis user would need a specific 3D printed foot

for each of their shoes that have a unique heel height (in other words, multiple prosthetic feet). Some

prosthesis users will be fine with this and some will not. The authors should consider mentioning this as a

possible limitation.”

An additional potential limitation to using the RECOVER ankle-feet system has been added in lines 210-217.

“The authors should add more references pertaining to current prosthetic feet used in clinic, alignment,

biomechanics of prosthesis users, the impacts alignment can have on the users, and rollover shape.”

We believe the additional information and citations provided in addressing the previous reviewer comments

also include this request and offer greater clarity to the reader.

---

## [Decision Letter · Decision Letter 1]

11 Mar 2025

PONE-D-24-25730R1Functional dynamic prosthesis alignment maintained across varying footwear using a modular prosthetic ankle-feet systemPLOS ONE

Dear Dr. Walker,

Thank you for submitting your manuscript to PLOS ONE. After careful consideration, we feel that it has merit but does not fully meet PLOS ONE’s publication criteria as it currently stands. Therefore, we invite you to submit a revised version of the manuscript that addresses the points raised during the review process.

We look forward to receiving your revised manuscript.

Kind regards,

Yih-Kuen Jan, PhD

Academic Editor

PLOS ONE

Journal Requirements:

Reviewers' comments:

Reviewer's Responses to Questions

**Comments to the Author**

1. If the authors have adequately addressed your comments raised in a previous round of review and you feel that this manuscript is now acceptable for publication, you may indicate that here to bypass the “Comments to the Author” section, enter your conflict of interest statement in the “Confidential to Editor” section, and submit your "Accept" recommendation.

Reviewer #1: All comments have been addressed

Reviewer #2: All comments have been addressed

2. Is the manuscript technically sound, and do the data support the conclusions?

Reviewer #1: Yes

Reviewer #2: Yes

3. Has the statistical analysis been performed appropriately and rigorously? 

Reviewer #1: Yes

Reviewer #2: Yes

4. Have the authors made all data underlying the findings in their manuscript fully available?

Reviewer #1: Yes

Reviewer #2: Yes

5. Is the manuscript presented in an intelligible fashion and written in standard English?

Reviewer #1: Yes

Reviewer #2: Yes

6. Review Comments to the Author

Reviewer #1: (No Response)

Reviewer #2: This revised manuscript offers more detail in areas lacking, specifically on RECOVER system and Figure 2.

I agree that evaluating frontal and transverse kinematics are warranted, as are moments and socket comfort across longer repeated durations. Roll over is an indices, however, moment of force may support your finding. Is this system a component which could be ordered? Would it fall under any current L-Codes?

7. PLOS authors have the option to publish the peer review history of their article (what does this mean? ). If published, this will include your full peer review and any attached files.

**Do you want your identity to be public for this peer review?** For information about this choice, including consent withdrawal, please see our Privacy Policy .

Reviewer #1: No

Reviewer #2: No

---

## [Author Response · Author response to Decision Letter 2]

10 Apr 2025

8 April 2025

Dear Editors-in-Chief,

We greatly appreciate the reviewers’ thoughtful evaluation of our manuscript. Please see our responses to the journal requirements and reviewer. No changes have been made to the manuscript content.

Journal Requirements:

All references have been reviewed and deemed up to date.

Reviewer 2:

“This revised manuscript offers more detail in areas lacking, specifically on RECOVER system and Figure 2.”

Thank you for this feedback.

“I agree that evaluating frontal and transverse kinematics are warranted, as are moments and socket comfort across longer repeated durations.”

We appreciate this feedback and will keep this in mind for future work.

“Roll over is an indices, however, moment of force may support your finding.”

We appreciate this feedback and will expand our future work to explore ankle torque and torque-angle relationships as well as roll-over shapes.

“Is this system a component which could be ordered? Would it fall under any current L-Codes?”

Our group is working on technology transfer to an industry partner. The most appropriate reimbursement coding is also being determined.

Sincerely,

Nicole Walker

Corresponding Author: Nicole Walker | Nicole.Walker6@va.gov | 612.467.3229

---

## [Editor Report · Decision Letter 2]

13 Apr 2025

Functional dynamic prosthesis alignment maintained across varying footwear using a modular prosthetic ankle-feet system

PONE-D-24-25730R2

Dear Dr. Walker,

We’re pleased to inform you that your manuscript has been judged scientifically suitable for publication and will be formally accepted for publication once it meets all outstanding technical requirements.

Kind regards,

Yih-Kuen Jan, PhD

Academic Editor

PLOS ONE
---

## [Editor Report · Acceptance letter]

PONE-D-24-25730R2

PLOS ONE

Dear Dr. Walker,

I'm pleased to inform you that your manuscript has been deemed suitable for publication in PLOS ONE. Congratulations! Your manuscript is now being handed over to our production team.

Kind regards,

on behalf of

Dr. Yih-Kuen Jan

Academic Editor

PLOS ONE